# THE THINKING SPECTRUM: AN EMPERICAL STUDY OF TUNABLE REASONING IN LLMS THROUGH MODEL MERGING

## ABSTRACT

The growing demand for large language models (LLMs) with tunable reasoning capabilities in many real-world applications highlights a critical need for methods that can efficiently produce a spectrum of models balancing reasoning depth and computational cost. Model merging has emerged as a promising, training-free technique to address this challenge by arithmetically combining the weights of a general-purpose model with a specialized reasoning model. While various merging techniques exist, their potential to create a spectrum of models with fine-grained control over reasoning abilities remains largely unexplored. This work presents a large-scale empirical study evaluating a range of model merging techniques across multiple reasoning benchmarks. We systematically vary merging strengths to construct accuracy-efficiency curves, providing the first comprehensive view of the tunable performance landscape. Our findings reveal that model merging offers an effective and controllable method for calibrating the trade-off between reasoning accuracy and token efficiency, even when parent models have highly divergent weight spaces. Crucially, we identify instances of Pareto Improvement, where a merged model achieves both higher accuracy and lower token consumption than one of its parents. Our study provides the first comprehensive analysis of this tunable space, offering practical guidelines for creating LLMs with specific reasoning profiles to meet diverse application demands. All our code and raw experimental results files are in https://anonymous.4open.science/r/ThinkingSpectrum-72E2.

## 1 INTRODUCTION

Large language models (LLMs) have become powerful, general-purpose tools, capable of addressing a wide array of tasks through a unified token generation process (Brown et al., 2020; Achiam et al., 2023; Liu et al., 2024). The field has seen the emergence of two distinct archetypes. On one end are "Thinking Models", such as DeepSeek-R1 (Jaech et al., 2024; Guo et al., 2025), which are optimized for maximum performance on complex tasks. They tend to generate extensive reasoning chains, consuming a significant number of tokens to achieve high accuracy. On the other end of the spectrum are "Direct Models," such as GPT-4o-mini (Hurst et al., 2024), which are not required to generate explicit intermediate reasoning tokens. This allows for rapid, low-latency responses, making them well-suited to simpler applications where speed is prioritized. However, many real-world applications demand a balance between these two extremes, requiring a spectrum of models with varying depths of reasoning. For instance, an LLM applied in secondary school mathematics education requires more reasoning depth than a fast conversational model, yet an IMO-level model would likely over-reason, incurring unnecessary latency and cost. This leads to a fundamental research question: how can we efficiently create a spectrum of LLMs with tunable reasoning abilities?

The motivation behind current research in efficient reasoning aligns with this goal: to create models that reduce computational cost, potentially with a controlled trade-off in reasoning capability. These include supervised fine-tuning (SFT) with shorter chain-of-thought (CoT) data (Kang et al., 2025; Munkhbat et al., 2025), incorporating output length constraints as an RL reward signal (Luo et al., 2025; Hou et al., 2025), and shifting reasoning from the token space to a more compact latent space (Shen et al., 2025; Saunshi et al., 2025). While effective, these methods necessitate additional,

often substantial, training resources, which makes them impractical in many resource-constrained scenarios and highlights the demand for a low-cost approach to achieving tunable reasoning.

In this context, model merging emerges as a promising, training-free alternative. Model merging typically combines the parameters of models specializing in different domains to create a single, more capable model. By treating deep thinking and direct response as distinct meta capabilities, model merging offers a path to craft models with a tailored trade-off between reasoning depth and token efficiency for specific applications. However, this is a promising yet underexplored direction. Existing studies (Team et al., 2025; Wu et al., 2025) have largely focused on producing a single merged model with enhanced accuracy or more concise reasoning, rather than systematically exploring the tunable accuracy-efficiency trade-offs. Consequently, a comprehensive understanding of how different merging techniques compare across this spectrum remains elusive, which is essential to guide the selection of optimal merging strategies for real-world applications where balancing performance and cost is critical.

This paper presents the first comprehensive empirical study focused on leveraging model merging for tunable reasoning. We evaluate seven representative model merging approaches, including Linear (Wortsman et al., 2022b), SLERP (Ilharco et al., 2023), TIES (Yadav et al., 2023), TWIN (Lu et al., 2024), EMR (Huang et al., 2024), DARE (Yu et al., 2024), and LORE (Liu et al., 2025b), across five diverse benchmarks: the reasoning-oriented AIME24, AIME25, and HMMT25; the multi-domain multiple-choice GPQA diamond (Rein et al., 2024); and the general Creative Writing benchmark (Paech, 2025). To assess scalability, our evaluation spans both a 4B dense LLM and a large 30B Mixture-of-Experts (MoE) model. By systematically sweeping the merging strength for each technique, we construct their complete accuracy-efficiency curves, providing the first detailed analysis of the performance trade-offs inherent to model merging.

Our findings are striking. We first validate that model merging effectively achieves tunable reasoning, even when parent models have highly divergent parameter spaces. Furthermore, we frequently observe *Pareto Improvements*, where a merged model surpasses its parent thinking model in both reasoning accuracy and token efficiency. What's more, we identify critical *phase changes* in performance, where subtle variations in merging strength within specific ranges lead to substantial shifts in model behavior. These phenomena can be explained by a single hypothesis: model merging is analogous to sampling intermediate checkpoints along the continuous post-training path that transforms a direct model into a thinking model. Ultimately, our study provides actionable insights and practical guidance for choosing and applying model merging techniques to optimize for both reasoning accuracy and token efficiency.

## 2 Preliminaries

### 2.1 Large Language Models and the Thinking Spectrum

Large language models have demonstrated remarkable general capabilities across tasks. In practice, the optimization of LLM capabilities typically follows two paradigms, constituting the endpoints of what we term the *Thinking Spectrum*:

- **Thinking Models** ($\theta_{\text{think}}$): These models are trained to generate exhaustive reasoning processes to maximize performance on complex problems. This is often achieved through supervised fine-tuning on datasets with detailed Chain-of-Thought (CoT) instructions or via reinforcement learning with verifiable rewards (Guo et al., 2025). Such models are trained to consistently generate a chain of thought, often engaging in further reflection, thereby producing longer token sequences. We denote these models as $\theta_{\text{think}}$.
- **Direct Models** ($\theta_{\text{direct}}$): These models tend to provide answers directly, without necessarily including explicit reasoning steps. They are suitable for simple application scenarios due to their relatively low latency and low computational cost. We denote these models as $\theta_{\text{direct}}$.

However, publicly available open-source models are often confined to these two extremes of the spectrum. Many real-world applications (e.g., educational tools, code assistants) require a balance between reasoning accuracy and computational cost. The central goal of this study is to explore methods for efficiently generating a spectrum of models with varying reasoning intensities that occupy the space between these two poles.

## 2.2 MODEL MERGING

Model merging refers to the arithmetic combination of weights from multiple source models directly in parameter space. The goal is to create a new model that inherits the capabilities of its constituents at zero additional training cost. The simplest method is linear weight averaging, which linearly interpolates between two models, $\theta_1$ and $\theta_2$.

$$\theta_{\text{merged}}(\lambda) = (1 - \lambda)\theta_1 + \lambda\theta_2, \quad \lambda \in [0, 1] \tag{1}$$

Task Arithmetic (Ilharco et al., 2023) provides a theoretical framework for such operations. Its core idea is that a specific capability acquired through fine-tuning can be represented as a separable and composable "task vector" ($\Delta\theta = \theta_{\text{task}} - \theta_{\text{base}}$). This perspective conceptualizes model capabilities as having a modular structure within the parameter space, where linear interpolation can be viewed as applying a scaled task vector to a base model. However, simple linear combination of parameters often fails due to destructive interference. To address this, more advanced merging algorithms have been proposed. For instance, TIES-Merging (Yadav et al., 2023) merges models by resolving sign conflicts in their parameter updates, while DARE (Yu et al., 2024) prunes the task vector before merging to eliminate redundancy and enhance robustness.

Prior research has primarily applied merging techniques to fuse models with complementary knowledge domains, such as programming and multilingual capabilities. This study, however, explores a novel application: merging models with different generative strategies, namely, a Thinking Model ($\theta_{\text{think}}$) and a Direct Model ($\theta_{\text{direct}}$), to construct a spectrum of models offering a tunable trade-off between cost and performance.

## 3 ANALYSIS

The effectiveness of model merging is theoretically underpinned by the principle of mode connectivity (Fort & Jastrzebski, 2019; Garipov et al., 2018b). This theory posits that multiple fine-tuned models (e.g., $\theta_1, \theta_2$) originating from the same pre-trained model, $\theta_{pre}$, co-exist within a broad, low-loss basin in the parameter space. Consequently, a low-loss path typically connects these solutions, allowing models interpolated along this path to retain high performance. This property provides the theoretical justification for simple merging methods like weighted averaging.

However, existing evidence for mode connectivity primarily stems from two scenarios: (1) models fine-tuned on distinct tasks or data distributions to acquire different domain knowledge (Ainsworth et al., 2022; Yadav et al., 2023), and (2) models that converge to different local optima due to varied random seeds (Wortsman et al., 2022a). In such cases, the divergence between models can be viewed as a local adjustment to the pre-trained capabilities, resulting in small parameter-wise differences.

The context of our study presents a more significant challenge. The disparity between $\theta_{think}$ and $\theta_{direct}$ is not a difference in knowledge but a fundamental divergence in their behavioral patterns, or computational strategy: a shift from a preference for immediate answers to one for deliberate reasoning. This strategic shift likely necessitates global, coordinated adjustments across the model's parameters, rather than minor and localized changes. Such a qualitative transformation could result in a substantial displacement in the parameter space, creating a distance between the models that far exceeds that observed between models specializing in different knowledge domains.

Therefore, in this new context, the risk of encountering a high-loss ridge along the interpolation path is significantly elevated. The effectiveness of merging methods that rely on the mode connectivity assumption can no longer be taken for granted. This uncertainty motivates us to first conduct an *a priori* evaluation of this risk before proceeding with large-scale merging experiments. Our subsequent analysis will quantitatively investigate the distance between $\theta_{think}$ and $\theta_{direct}$ in the parameter space by analyzing the magnitude and distribution of their parameter differences. This allows us to make a preliminary judgment on the potential viability of simple merging methods.

We analyze the parameter differences between two pairs of "thinking" and "direct" models: (1) Qwen3-4B-Thinking-2507 and Qwen3-4B-Instruct-2507, and (2) Qwen3-30B-A3B-Thinking-2507 and Qwen3-30B-A3B-Instruct-2507. Our analysis yields the following key findings:

- **Significant Parameter Differences.** We plot the distribution of parameter differences for both model pairs and compare them against a typical model pair from Yu et al. (2024) (NeuralBeagle14-

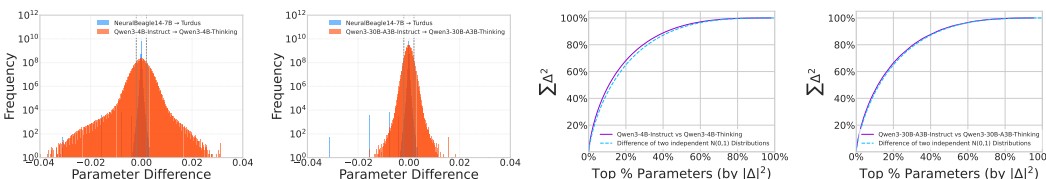

(a) Distribution of $\Delta\theta$, Qwen3-4B. (b) Distribution of $\Delta\theta$, Qwen3-30B. (c) Cumulative distribution of $\Delta\theta^2$, Qwen3-4B. (d) Cumulative distribution of $\Delta\theta^2$, Qwen3-30B.

Figure 1: Analysis of parameter differences ($\Delta\theta = \theta_{think} - \theta_{direct}$). Figures (a) and (b) show the distribution of parameter-wise differences. Figures (c) and (d) show the cumulative distribution of the squared parameter differences ($\Delta\theta_i^2$), compared against a reference distribution.

7B and Turdus), whose merger achieved the top rank on the OpenLLM Leaderboard at the time. In Figures 1a and 1b, the dashed vertical lines represent the $\pm 0.002$ threshold. This range was noted by (Yu et al., 2024) as containing the vast majority of parameter differences in their successful merge. In stark contrast, we observe that the parameter differences between our thinking and direct models are substantially larger and more widely distributed, far exceeding this range. This indicates a much greater parameter-wise divergence than what is typically seen between models with different domain specializations.

- **Large Global Distance.** To quantify the overall distance between the models, we calculate the relative distance, defined as the ratio of the L2 norm of the parameter delta to the L2 norm of the direct model's parameters, i.e., $\|\theta_{think} - \theta_{direct}\|_2 / \|\theta_{direct}\|_2$. The two successful merging examples in (Yu et al., 2024) (NeuralBeagle14-7B vs. Turdus and WildMarcoroni-Variant1-7B vs. WestSeverus-7B-DPO-v2) exhibit small relative distances of 0.5486% and 0.6350%, respectively. In contrast, the model pairs we analyze show significantly larger distances: 7.9048% for the 4B models and 3.7816% for the 30B models. This finding substantiates our hypothesis that the shift from a direct to a reasoning computational strategy corresponds to a much larger displacement in parameter space than that of fine-tuning for different knowledge domains.

- **Dense Differences.** As shown in Figures 1c and 1d, we analyzed the cumulative distribution of the squared parameter differences. For both the 4B and 30B models, the shape of this distribution closely aligns with that of the difference between two independent, identically distributed Gaussian variables. This suggests that the parameter delta lacks non-trivial sparsity and is instead relatively dense. This density implies that the "deep thinking" capability is not a localized, plug-and-play module but rather the result of a global, coordinated adjustment across the network. To achieve this strategic shift, a vast number of parameters throughout the model have undergone subtle, interconnected modifications.

What's more, previous discussions on the sparsity of parameter differences have considered not only the inherent sparsity of the delta but also whether the complete delta can be effectively replaced by a sparse subset. To answer this question, we apply DARE by randomly pruning 99%, 90%, and even 50% of the parameter differences between our models and their corresponding Instruct/Thinking versions, rescaling the remaining differences accordingly. However, unlike the fine-tuned models reported by Yu et al. (2024), which maintained performance even when 99% of the parameter delta was pruned, a mere 50% sparsification causes catastrophic damage to our models, resulting in a near-complete loss of language capabilities. The same observation holds when using the corresponding pre-trained model as the base. This indicates that the difference between the Instruct and Thinking models is inherently non-sparse and cannot be easily replaced by a sparse approximation.

Collectively, these analyses demonstrate that the parameter-level divergence between the thinking and direct models is substantial. This raises considerable doubt regarding the effectiveness of simple merging methods that presuppose mode connectivity. Nevertheless, we proceed to empirically evaluate the performance of various model merging techniques in the following sections.

## 4 EXPERIMENTAL SETUP

To investigate the feasibility of generating models with tunable reasoning capabilities along a "thinking spectrum," we systematically evaluated a variety of model merging algorithms across different model architectures and parameter scales.

The model pairs used in our study are: Qwen3-4B-Instruct-2507 (the fast-response model, $\theta_{\text{direct}}$) and Qwen3-4B-Thinking-2507 (the deep-thinking model, $\theta_{\text{think}}$); Qwen3-30B-A3B-Instruct-2507 ($\theta_{\text{direct}}$) and Qwen3-30B-A3B-Thinking-2507 ($\theta_{\text{think}}$).

We employed a range of established model merging techniques:

- **Weighted Average**: This method performs a weighted average of the two models' parameters, directly interpolating in the parameter space by adjusting a weight coefficient $\lambda$ according to the formula $\theta_{\text{merged}}(\lambda) = (1 - \lambda)\theta_{\text{direct}} + \lambda\theta_{\text{think}}$.
- **Spherical Linear Interpolation (SLERP)**: To maintain the norm of the model parameters during interpolation, the merged model is computed as $\theta_{\text{merged}}(t) = \frac{\sin((1-t)\Omega)}{\sin(\Omega)}\theta_{\text{direct}} + \frac{\sin(t\Omega)}{\sin(\Omega)}\theta_{\text{think}}$, where $\Omega$ is the angle between $\theta_{\text{direct}}$ and $\theta_{\text{think}}$.
- **DARE (Drop And REscale)**: A task vector-based pruning method that randomly drops a fraction of parameters from the task vector (e.g., $\theta_{\text{think}} - \theta_{\text{base}}$) and rescales the remainder to mitigate parameter conflicts while preserving core abilities.
- **TIES-Merging (Trim, Elect Sign & Merge)**: This method also operates on task vectors, but it reduces interference by trimming low-magnitude parameters, resolving sign conflicts among parameter updates, and finally merging the aligned vectors.
- **EMR-Merging (Elect, Mask & Rescale)**: This approach first elects a unified parameter model and then generates lightweight, task-specific directional masks and magnitude rescalers for each source model to align and reconstruct them.
- **LORE-Merging (Low-Rank Estimation)**: This method formulates the model merging problem as a low-rank estimation task, which is solved via singular value thresholding (SVT) to identify and fuse the core low-rank structures of task vectors.
- **TWIN-Merging**: This method decomposes model knowledge into shared and exclusive components. In our static setting, it computes a shared model first, then extracts, sparsifies, and merges the exclusive knowledge vectors.

Furthermore, to test the robustness of model merging in our specific context, we designed three arbitrary custom fusion strategies:

- **Top-K Replacement**: This strategy identifies the top k% of parameters with the largest absolute difference between the two models and directly replaces the parameters in the fast-response model ($\theta_{\text{direct}}$) with those from the deep-thinking model ($\theta_{\text{think}}$).
- **Top-K Difference Averaging**: This strategy identifies the top k% of parameters with the largest absolute difference and computes their average, while all other parameters are retained from the fast-response model ($\theta_{\text{direct}}$).
- **Global Average with Top-K Override**: This approach first computes the simple average of all parameters across both models. It then identifies the top k% of parameters with the largest original difference and overwrites their averaged values with the original parameters from the deep-thinking model ($\theta_{\text{think}}$).

Through this diverse set of methods, we aim to comprehensively evaluate the potential and limitations of model merging for constructing a spectrum of reasoning capabilities.

## 5 RESULTS AND FINDINGS

Our comprehensive experimental results are presented in Figures 2 through 6. We detail our key findings below. Shaded regions in the figures indicate 90% confidence intervals for the reasoning benchmarks (AIME24, AIME25, HMMT25).

**Model merging enables tunable reasoning despite large parameter distances.** Contrary to the concerns raised by the significant parameter-space divergence analyzed in Section 3, our results demonstrate that even simple interpolation methods (Weighted Average and SLERP) effectively

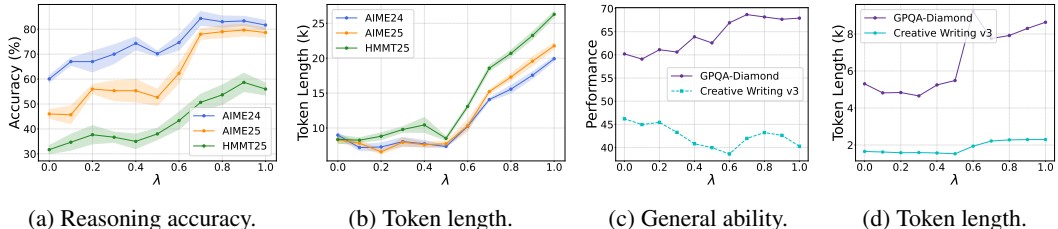

(a) Reasoning accuracy.     (b) Token length.     (c) General ability.     (d) Token length.

Figure 2: Performance and token consumption of Qwen3-4B models merged using Weighted Average across varying merging strengths ($\lambda$).

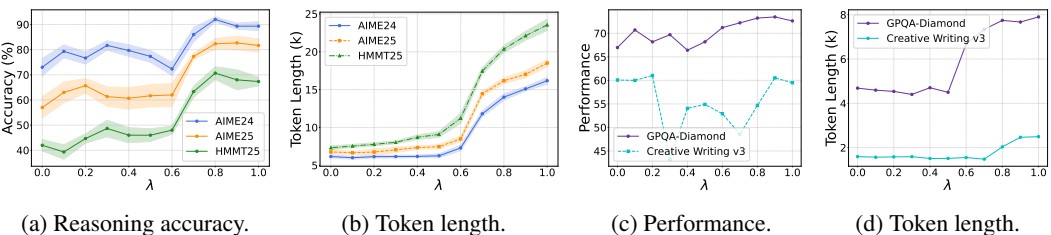

(a) Reasoning accuracy.     (b) Token length.     (c) Performance.     (d) Token length.

Figure 3: Performance and token consumption of Qwen3-30B models merged using Weighted Average across varying merging strengths ($\lambda$).

create a spectrum of models balancing reasoning accuracy and efficiency. As shown in Figures 2 through 5, increasing the weight of the thinking model ($\theta_{\text{think}}$) generally leads to increased token consumption and improved accuracy on reasoning tasks. Crucially, the models interpolated along this path maintain reasonable performance and do not exhibit catastrophic failure (i.e., a "high-loss ridge"). This surprising result suggests that $\theta_{\text{direct}}$ and $\theta_{\text{think}}$, despite their fundamental behavioral differences and large parameter distance, still reside within a broad, connected low-loss basin.

Furthermore, the general capabilities of the merged models remain relatively stable. Performance on the multidisciplinary GPQA benchmark is largely preserved (see subfigures (c) in Figures 2-5). While a decline is observed in Creative Writing at certain merging strengths, qualitative analysis reveals this is primarily due to formatting inconsistencies arising from the mixture of direct and reasoning response styles, rather than a degradation of inherent writing quality under human evaluation.

**Merged models can achieve Pareto improvements over parent models.** A significant finding is the frequent occurrence of Pareto improvements, where a merged model surpasses the original thinking model ($\theta_{\text{think}}$) in both reasoning accuracy and efficiency (lower token consumption). For instance, the Qwen3-4B model merged with Weighted Average at $\lambda = 0.8$ (Figure 2) and the Qwen3-30B model at $\lambda = 0.7$ (Figure 3) both exhibit this phenomenon. This is further illustrated in the accuracy-efficiency curves (Figure 6), where numerous merged configurations reside in the upper-left quadrant relative to $\theta_{\text{think}}$. This demonstrates that model merging is not merely a tool for creating trade-offs but also a viable method for discovering models that are faster and more accurate than their specialized parents for potential target applications.

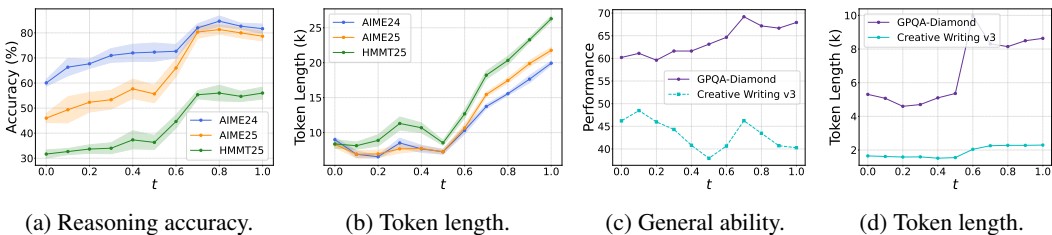

(a) Reasoning accuracy.     (b) Token length.     (c) General ability.     (d) Token length.

Figure 4: Performance and token consumption of Qwen3-4B models merged using SLERP across varying interpolation strengths ($t$).

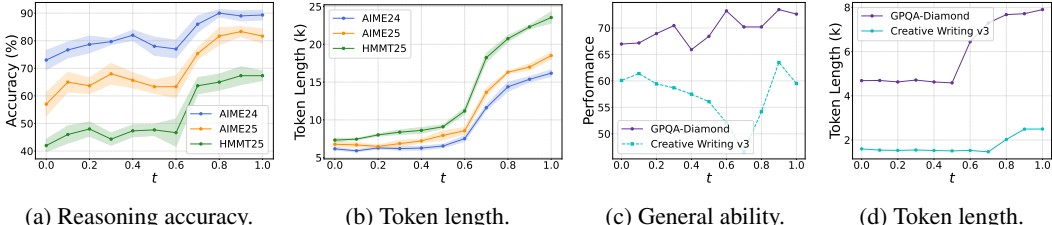

(a) Reasoning accuracy.  (b) Token length.  (c) General ability.  (d) Token length.

Figure 5: Performance and token consumption of Qwen3-30B models merged using SLERP across varying interpolation strengths ($t$).

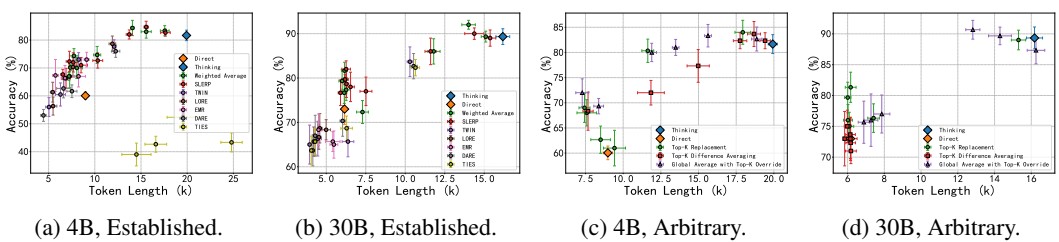

(a) 4B, Established.  (b) 30B, Established.  (c) 4B, Arbitrary.  (d) 30B, Arbitrary.

Figure 6: Accuracy-Efficiency trade-off curves (Pareto fronts) for various merging methods on the AIME24 benchmark. Established methods (a, b) are from prior work, while Arbitrary methods are designed for this study. The Direct ($\theta_{\text{direct}}$) and Thinking ($\theta_{\text{think}}$) models are marked for reference. Points higher and further left indicate better trade-offs.

**Reasoning behavior exhibits non-linear phase changes.** The transition from direct response to deep thinking is not linear with respect to the merging weight. As observed across Figures 2 through 5, both reasoning accuracy and token consumption change slowly at lower merging weights. However, there is a critical region, typically around $\lambda/t \in [0.6, 0.7]$, where performance and token usage increase rapidly. This "phase change" or emergent behavior suggests that the activation of complex reasoning pathways requires a critical threshold of parameter adjustments, indicating a non-trivial transition in the model's computational strategy even within a connected low-loss basin.

**Established merging methods yield similar accuracy-efficiency trade-offs.** When comparing the various established model merging techniques, we find that they generally fall along a similar Pareto front. As shown in Figures 6a and 6b, while minor variations exist, no single method consistently dominates the others across the spectrum of trade-offs, especially when considering the 90% confidence intervals. The low performance of TIES in the 4B setting might be attributed to the specific hyperparameter sensitivity or the challenges of resolving sign conflicts in this highly divergent context for smaller models. Overall, the specific choice of merging algorithm appears less critical than the ability to tune the merging strength.

**The merging process is highly robust, even to arbitrary fusion strategies.** To further test the stability of the interpolation path, we evaluated three arbitrary merging strategies (Top-K Replacement, Top-K Difference Averaging, and Global Average with Top-K Override). Surprisingly, as shown in Figures 6c and 6d, these methods, despite lacking theoretical motivation, also produce functional models that reside near the Pareto front established by the principled methods. They do not lead to model collapse. This remarkable robustness reinforces the observation that the parameter space between the direct and thinking models is highly permissive to interpolation.

## 6   DISCUSSIONS

In this section, we delve deeper into the observed phase transition in reasoning ability, and propose a hypothesis for the effectiveness of model merging. We also outline several unresolved questions that stem from our findings.

We try to investigate the "phase change" in reasoning capabilities more closely. Our previous experiments indicate a rapid performance gain as the linear interpolation weight, $\lambda$, increases from 0.6 to

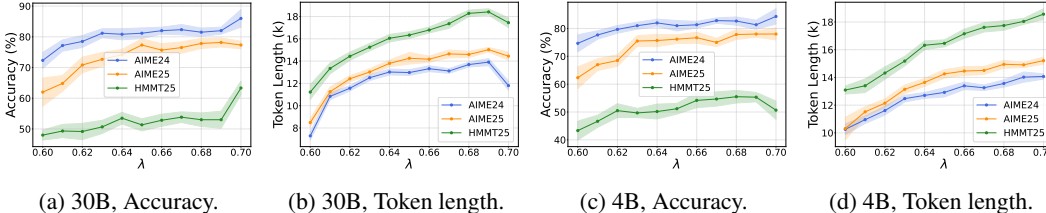

(a) 30B, Accuracy.     (b) 30B, Token length.     (c) 4B, Accuracy.     (d) 4B, Token length.

Figure 7: Finer-grained analysis of the phase transition for Weighted Average merging within the critical $\lambda \in [0.6, 0.7]$ interval. Panels (a, b) show results for the Qwen3-30B models, and (c, d) for the Qwen3-4B models. The performance jump occurs at a higher $\lambda$ for more difficult benchmarks (AIME24 < AIME25 < HMMT25), analogous to emergent abilities.

0.7. To understand this transition, we conduct a new parameter sweep within this critical interval, sampling from $\lambda = 0.6$ to $\lambda = 0.7$ at intervals of 0.01. To ensure rigor our limited computational resources, each experiment is repeated 20 times. The results for both the Qwen3-4B and Qwen3-30B model pairs are presented in Figure 7.

The results reveal that the primary interval of performance increase varies across benchmarks of differing difficulty. As benchmark complexity increases, this critical interval appears later (i.e., requires a higher $\lambda$). For instance, with the 30B model, the main performance gain on AIME24 occurs between $\lambda = 0.60$ and $\lambda = 0.63$. For the more challenging AIME25, the gain is concentrated between $0.61$ and $0.65$, while for the most difficult benchmark, HMMT25, significant improvement only begins around $\lambda = 0.69$. A similar trend is observable for the 4B model. This behavior is strikingly analogous to the well-documented phenomenon of **emergent abilities** in LLMs, where performance on a given task sharply increases within a specific range of training compute, and this emergence occurs later for more complex tasks (Wei et al., 2022; Snell et al., 2024).

Based on this analogy, we hypothesize that: **simple model merging can approximate the process of sampling intermediate checkpoints along a continuous post-training trajectory that transforms the direct model into the thinking model.**[1]

This hypothesis provides a compelling explanation for our key findings. The **emergence** of reasoning ability during merging mirrors the emergence during training. The existence of **Pareto improvements** can be explained by the thinking model being "over-trained" for our specific benchmarks. Specifically, its performance may have saturated midway through its training, but continued training biased it towards generating longer, more costly token sequences. Merging with the direct model effectively *rolls back* the model to a checkpoint near this saturation point, achieving similar accuracy with higher efficiency. This perspective also explains the **robustness** of the merging process. Because our method approximates sampling a checkpoint from a stable training trajectory, the resulting merged models are consistently functional and avoid catastrophic failure. However, as was also observed, they may introduce minor artifacts, such as formatting inconsistencies, which manifest as mixed response styles or the inclusion of stray reasoning tags like <think>.

This insight brings us back to the initial motivation of our work. If an application requires reasoning but not at the highest possible intensity, a moderately trained model would suffice. However, training such a checkpoint is often infeasible in low-resource scenarios. In such cases, we initially proposed model merging as a training-free *alternative* to achieve the desired accuracy-cost trade-off. We now find that **these two approaches are more analogous than we had imagined**; this gives us stronger reason to believe that model merging is an excellent substitute for training, offering a highly effective solution for creating a model with well-calibrated reasoning depth.

We must acknowledge, however, that our hypothesis currently lacks a rigorous mathematical foundation. This leads to several open questions for future work:

1. Can a formal mathematical framework be developed to explain the effectiveness of model merging in this non-typical scenario of interpolating computational strategies?
2. Do methods exist that can yield a significantly better Pareto front than simple linear interpolation?

---

[1]While this represents a plausible training trajectory, the actual method used by the model creators differs.

3. For a given task, is it possible to predict the optimal merging weight without resorting to an expensive empirical search?

We hope our work will inspire future research into these important questions.

## 7 RELATED WORKS

**Efficient Reasoning.** The substantial computational overhead associated with the "slow-thinking" models has spurred a significant research effort toward efficient reasoning (Feng et al., 2025). These efforts can be broadly categorized into three main directions. The first focuses on compressing lengthy CoTs into more concise reasoning chains. This is often achieved through training-based methods, such as reinforcement learning with length penalties (Luo et al., 2025; Hou et al., 2025) or supervised fine-tuning on shorter CoT data (Ma et al., 2025; Xia et al., 2025). A second direction aims to develop compact yet powerful reasoning models through techniques like knowledge distillation (Gu et al., 2024; Liao et al., 2024), quantization and pruning (Liu et al., 2025a; Zhang et al., 2025), as well as RL on smaller models (Zeng et al., 2025). The third centers on designing more efficient decoding strategies (Lin et al., 2025; Xu et al., 2025; Wang et al., 2025), such as speculative rejection (Sun et al., 2024) and parallel decoding (Ding et al., 2025; Jin et al., 2024), to accelerate inference without altering the model's core reasoning path. While these approaches have shown promise, they often require additional training or complex modifications to the inference process. Our work explores model merging as an orthogonal, training-free alternative that can achieve tunable reasoning efficiency in a low-cost way.

**Model Merging.** Model merging offers a training-free paradigm for combining the capabilities of multiple specialized models into a single checkpoint (Wan et al., 2024; Lu et al., 2024; Huang et al., 2024; Deep et al., 2024; Davari & Belilovsky, 2024). The foundational concept involves arithmetically averaging the parameters of models fine-tuned from a common initialization, assuming that such models share a connected, low-error basin in the loss landscape (Garipov et al., 2018a; Wortsman et al., 2022b). Building on this, Task Arithmetic (Ilharco et al., 2023) introduced the concept of "task vectors" characterizing the difference between fine-tuned and base model weights which enables more sophisticated and semantically meaningful combinations. A key challenge in merging is mitigating "parameter interference," where conflicting task vectors can degrade performance. To address this, methods like TIES-merging (Yadav et al., 2023) and DARE (Yu et al., 2024) introduce sparsity, selectively combining only the most significant parameter changes to resolve conflicts. Beyond linear and sparse combinations, other methods explore non-linear interpolation paths like SLERP (Goddard et al., 2024) or employ low-rank approximations to distill task-specific knowledge more robustly (Liu et al., 2025b). While these techniques have proven effective for creating powerful multitask models, their systematic application for creating a tunable spectrum of reasoning abilities remains unexamined.

## 8 CONCLUSION

This work presents the first comprehensive empirical study demonstrating the potential of model merging as a training-free method to generate a spectrum of LLMs with tunable reasoning capabilities. By arithmetically combining general-purpose (direct) and specialized (thinking) models, we have shown that it is possible to achieve fine-grained control over the trade-off between reasoning accuracy and computational efficiency. Our extensive evaluation across diverse merging techniques and model scales reveals that this approach is surprisingly effective, even when parent models exhibit significant divergence in their parameter spaces.

Crucially, our findings highlight the frequent occurrence of Pareto improvements, where merged models surpass their thinking parents in both accuracy and efficiency. Furthermore, we characterize the non-linear dynamics of reasoning emergence, observing distinct phase changes during interpolation. We hypothesize that these phenomena occur because model merging approximates sampling from a continuous training trajectory between the direct and thinking models. Our study establishes a strong foundation and provides practical guidelines for efficiently creating LLMs tailored to specific computational budgets and diverse application demands, paving the way for more accessible and optimized deployment of advanced reasoning models.

REPRODUCIBILITY STATEMENT

To ensure the reproducibility of our findings, we make our code, configuration files, evaluation scripts, and **the settings and raw LLM outputs for every single experiment** publicly available anonymously. All experiments were conducted using publicly available models from the Hugging Face Hub and standard reasoning benchmarks, including AIME24, AIME25, HMMT25, GPQA diamond, and the Creative Writing benchmark.

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

# A APPENDIX

## A.1 USE OF LLMS

We use LLMs to polish some paragraphs of the manuscript. All the research ideas and designs are conceived by the authors.

## A.2 IMPLEMENTATION DETAILS OF MERGING METHODS

All model merging operations in this study are performed between a deep-thinking model, $\theta_{\text{think}}$, and a fast-response model, $\theta_{\text{direct}}$. For algorithms requiring a base model, $\theta_{\text{base}}$, we consistently use the corresponding Qwen3-4B or Qwen3-30B-A3B model. This choice is predicated on the assumption that keeping the task vectors (i.e., the parameter delta from the base to the specialized models) relatively small will ground the merge process and prevent the resulting model from drifting too far from a pretrained foundation. For methods that require specifying a parameter retention or dropping ratio, we uniformly set the drop rate to 0.2 (implying a density or retention ratio of 0.8).

- **Weighted Average.** This method serves as the most fundamental baseline, directly combining the two source models in the parameter space. For each corresponding parameter tensor in $\theta_{\text{direct}}$ and $\theta_{\text{think}}$, the merged tensor is computed as a simple weighted average. The entire merged model is defined by the interpolation coefficient $\lambda \in [0, 1]$ as follows:

$$\theta_{\text{merged}}(\lambda) = (1 - \lambda)\theta_{\text{direct}} + \lambda\theta_{\text{think}}$$

  When $\lambda = 0$, the merged model is identical to $\theta_{\text{direct}}$, and when $\lambda = 1$, it is identical to $\theta_{\text{think}}$.

- **Spherical Linear Interpolation (SLERP)** Spherical Linear Interpolation aims to provide a smooth transition along the geodesic path in the parameter space, preserving the geometric properties of the weights. For each parameter tensor $v_0$ from $\theta_{\text{direct}}$ and its corresponding tensor $v_1$ from $\theta_{\text{think}}$, our implementation computes the merged tensor $v_{\text{merged}}$ using the standard SLERP formula. The angle $\theta$ between the two tensor vectors is first calculated as:

$$\theta = \arccos\left(\frac{v_0 \cdot v_1}{\|v_0\|\|v_1\|}\right)$$

  The merged tensor is then computed using an interpolation coefficient $t \in [0, 1]$, which corresponds to the weight $\lambda$ in our experiments:

$$v_{\text{merged}} = \text{SLERP}(t; v_0, v_1) = \frac{\sin((1-t)\theta)}{\sin(\theta)}v_0 + \frac{\sin(t\theta)}{\sin(\theta)}v_1$$

  For numerical stability, our implementation defaults to linear interpolation when the two tensors are nearly collinear (i.e., their dot product is close to 1).

- **DARE (Drop And REscale).** The DARE method mitigates parameter interference by randomly sparsifying task vectors. Our implementation operates on task vectors relative to a base model, $\theta_{\text{base}}$. It first computes the task vectors for both the fast-response and deep-thinking models: $\Delta\theta_{\text{direct}} = \theta_{\text{direct}} - \theta_{\text{base}}$ and $\Delta\theta_{\text{think}} = \theta_{\text{think}} - \theta_{\text{base}}$. Each of these task vectors is then processed independently through the DARE procedure: a fraction of its parameters are randomly set to zero with a probability $p$ (the drop rate), and the remaining non-zero parameters are rescaled by a factor of $1/(1-p)$. This results in two sparse and rescaled task vectors, $\Delta\theta'_{\text{direct}}$ and $\Delta\theta'_{\text{think}}$. Finally, these processed vectors are linearly combined using the weight $\lambda$ and added back to the base model to produce the merged model:

$$\theta_{\text{merged}} = \theta_{\text{base}} + ((1 - \lambda) \cdot \Delta\theta'_{\text{direct}} + \lambda \cdot \Delta\theta'_{\text{think}})$$

- **TIES-Merging.** This method resolves interference between task vectors via a three-step "Trim, Elect Sign, and Merge" process. Our implementation begins by computing the task vectors $\Delta\theta_{\text{direct}}$ and $\Delta\theta_{\text{think}}$. The **Trim** step performs a local pruning on each parameter tensor within the task vectors, retaining only the parameters with the highest magnitudes, as determined by a density hyperparameter. In the **Elect Sign** step, a consensus sign for each parameter position is determined via a weighted vote, using weights $[1 - \lambda, \lambda]$ for the two trimmed task vectors. Finally, the **Merge** step combines only the parameters from each vector that align with the consensus sign, normalized by their respective weights, to create a final merged task vector, $\Delta\theta_{\text{merged}}$, which is then added to the base model $\theta_{\text{base}}$.

- **EMR-Merging.** EMR-Merging operates via an "Elect, Mask, and Rescale" mechanism. The implementation first computes the task vectors $\Delta\theta_{\text{direct}}$ and $\Delta\theta_{\text{think}}$. The **Elect** step establishes a dominant direction (sign) based on the average of the two task vectors and constructs a unified vector by selecting the maximum parameter magnitude along this direction. Subsequently, the **Mask & Rescale** step generates a binary mask for each original task vector (identifying parameters aligned with the dominant direction) and a scaling factor (to preserve the original vector's average magnitude). The final model is constructed by adding a weighted sum (with weights $[1-\lambda, \lambda]$) of the two reconstructed task vectors to the base model.

- **LORE-Merging.** LORE-Merging frames model merging as a low-rank estimation problem and does not rely on a predefined base model. Instead, it solves an optimization problem to find an approximate shared base model, $\theta_0$. Given the input models $\theta_{\text{direct}}$ and $\theta_{\text{think}}$, the algorithm iteratively updates $\theta_0$ and two corresponding low-rank task vectors, $\delta_{\text{direct}}$ and $\delta_{\text{think}}$, using a coordinate descent method. After the optimization converges, the learned low-rank task vectors are combined using weights $[1-\lambda, \lambda]$ and added to the approximated base model $\theta_0$ to form the final merged model.

- **TWIN-Merging.** This method is designed to separate knowledge into shared and exclusive components. In our static fusion scenario, a shared model, $\theta_{\text{shared}}$, is first created by averaging the two task vectors ($\Delta\theta_{\text{direct}}$ and $\Delta\theta_{\text{think}}$) and adding the result to the base model $\theta_{\text{base}}$. Exclusive knowledge vectors are then extracted by computing the difference between each full model and the shared model ($v_{\text{direct}} = \theta_{\text{direct}} - \theta_{\text{shared}}$ and $v_{\text{think}} = \theta_{\text{think}} - \theta_{\text{shared}}$). These exclusive vectors are sparsified according to a mask rate, linearly combined using the weight $\lambda$, and finally added back to the shared model $\theta_{\text{shared}}$.

### A.2.1 CUSTOM MERGING STRATEGIES

To probe the robustness of the merging process, we implemented three bespoke strategies:

- **Top-K Replacement.**: This strategy identifies the top k% of parameters with the largest absolute difference between $\theta_{\text{direct}}$ and $\theta_{\text{think}}$. It then directly overwrites the values at these positions in $\theta_{\text{direct}}$ with the corresponding values from $\theta_{\text{think}}$, leaving all other parameters unchanged.

- **Top-K Difference Averaging.**: This approach identifies the k% of parameters with the largest absolute difference between the two models. At these positions, the parameters are replaced by the average of the values from both models. All other parameters, where the difference is the k% largest, are retained from $\theta_{\text{direct}}$.

- **Global Average with Top-K Override.**: This strategy first computes a global average of all parameters from $\theta_{\text{direct}}$ and $\theta_{\text{think}}$. It then identifies the top k% of parameter positions that had the largest original difference and overwrites the averaged values at these specific positions with the original values from $\theta_{\text{think}}$. This selectively injects critical parameters from the thinking model into a generally averaged model.

