# OpenReview forum: "The Thinking Spectrum: An Empirical Study of Tunable Reasoning in LLMs Through Model Merging"
_ICLR.cc/2026/Conference — ICLR 2026 Conference Withdrawn Submission_

### Official Review · Reviewer_RF2r · 2025-10-22

**Soundness:** 1
**Presentation:** 1
**Contribution:** 2
**Rating:** 2
**Confidence:** 3

**Summary:**

This work conducts an empirical study on using model merging to create LLMs with tunable reasoning capabilities. It identifies Pareto Improvements where merged models achieve both higher accuracy and lower token consumption than their parent models.

**Strengths:**

1. The paper systematically evaluates 7 merging techniques across 5 diverse benchmarks (AIME24, AIME25, HMMT25, GPQA, Creative Writing) and two model scales (4B dense and 30B MoE), constructing complete accuracy-efficiency curves.

**Weaknesses:**

1. The comparison between NeuralBeagle14-7B and Turdus (Section 3) could be strengthened. Their small parameter distance (0.5%) may primarily reflect alignment strategy differences rather than capability shifts. Additionally, comparing this pair with the Qwen series introduces potential confounds from different architectures and training strategies, which could make interpretation challenging. The study would benefit from more carefully matched baselines or an explicit discussion of these confounding factors.
2. The paper would be strengthened by a deeper analysis of why merging works effectively. Given that simple linear interpolation yields competitive or superior results compared to sophisticated algorithms (TIES, DARE, LORE), it would be valuable to discuss whether more scenarios might benefit from complex methods.
3. The paper over-relies on parameter distance as a connectivity proxy without directly measuring loss landscapes along interpolation paths, which would provide stronger evidence for the checkpoint sampling hypothesis.
4. Lack of Qualitative Analysis: How does the merged model actually solve problems differently from its parents? Without examining reasoning chains, the paper only shows metric improvements but doesn't explain how the model's thinking changes( through shorter reasoning, different strategies, or skipping steps ?).

**Questions:**

1. How do results generalize to other model families beyond Qwen?

---

### Official Review · Reviewer_godS · 2025-10-27

**Soundness:** 3
**Presentation:** 3
**Contribution:** 3
**Rating:** 4
**Confidence:** 3

**Summary:**

The paper investigates model merging as a training-free method for producing LLMs with tunable reasoning depth, positioned between “direct” (fast, shallow reasoning) and “thinking” (slow, chain-of-thought-heavy) models.
Using Qwen3-4B and Qwen3-30B models, it merges a reasoning-optimized model (θ_think) and a direct model (θ_direct) via various algorithms (Weighted Average, SLERP, DARE, TIES, EMR, LORE, TWIN, etc.).
Across multiple benchmarks (AIME24/25, HMMT25, GPQA-Diamond, Creative Writing), the authors analyze accuracy–efficiency trade-offs,

**Strengths:**

1. Introduces the “Thinking Spectrum” as a continuous axis between direct and thinking models, an appealing abstraction for reasoning control.
2. Evaluates seven established merging algorithms plus custom heuristics across multiple datasets and model sizes.
3. Demonstrates meaningful control of reasoning depth without any retraining, relevant for resource-limited applications.

**Weaknesses:**

1. The central hypothesis (“merging ≈ checkpoint sampling”) remains purely speculative with no mathematical support or gradient-path analysis.
2. Claims that direct and thinking models share a connected basin are asserted but not proven (no loss-surface probing or interpolation-loss plots).
3. Heavy reliance on mathematical reasoning datasets; generalization to broader reasoning types (commonsense, multi-hop, scientific) is untested.
4. No systematic study of the impact of architecture scale, base model family, or tokenizer differences on merge viability.
5. Statistical analyses are mostly descriptive; lacks formal significance testing, variance reporting across seeds, or ablation of random initialization.
6. “Token length” is used as proxy for cost, but latency or FLOPs would be more accurate measures of efficiency.
7. No direct quantitative comparison versus SFT-on-short-CoT, RLHF-pruning, or distillation baselines.

**Questions:**

-

---

### Official Review · Reviewer_WYxG · 2025-11-01

**Soundness:** 3
**Presentation:** 2
**Contribution:** 2
**Rating:** 2
**Confidence:** 3

**Summary:**

This paper shows an empirical study of model merging between base and thinking models for reasoning tasks. The authors find that the merged models can surpass their parent thinking model in both reasoning accuracy and token efficiency.

**Strengths:**

1. The paper shows some findings of model merging between base and thinking models, such as the merged model can achieve comparable reasoning accuracy with less token consumption.
2. The study is exceptionally thorough, evaluating seven merging techniques (e.g., Linear, SLERP, TIES, DARE) and three custom strategies across two model scales (4B and 30B parameters) and five diverse benchmarks.

**Weaknesses:**

1. The experiments are confined to the Qwen family of models.
2. The study briefly mentions artifacts like formatting inconsistencies and stray reasoning tags (<think>) in merged model outputs. These issues related to output stability, controllability, and potential failure modes are critical for real-world deployment but are not analyzed in depth.
3. The paper acknowledges the lack of a formal theory to explain why merging works so well despite the large parameter-space divergence between the parent models.

**Questions:**

1. Can a formal mathematical framework be developed to explain the surprising effectiveness of model merging?
2. How does model merging compare to model distillation?

---

### Note · Authors · 2025-12-03

**Comment:**

We have decided to withdraw the paper to conduct further experiments, incorporate the reviewers' feedback and improve the manuscript for a future submission. We thank all reviewers for their reviews.

**Withdrawal Confirmation:**

I have read and agree with the venue's withdrawal policy on behalf of myself and my co-authors.